# Magnitude of malnutrition and its associated factors among pediatric cancer patients on chemotherapy at oncology centers in Addis Ababa, Ethiopia, 2024

**Habtamu Wondmagegn Atlaw**[1]*, **Edlework Wondmagegn Atlaw**[2], **Biniyam Demisse Andarge**[3], **Sayih Mehari Degualem**[3], **Tsegazeab Ayele Meshesha**[1], **Maycas Gembe**[4], **Habtamu Esubalew Bezie**[5]

1 Department of Anatomy, College of Medicine and Health Sciences, Arba Minch University, Arba Minch, Ethiopia, 2 Department of Oncology, SPHMMC, Addis Ababa, Ethiopia, 3 Department of Nursing, College of Medicine and Health Sciences, Arba Minch University, Arba Minch, Ethiopia, 4 Department of Epidemiology and Biostatistics, School of Public Health, Yekatit 12 Hospital Medical College, Addis Ababa, Ethiopia, 5 Department of Public Health, College of Medicine and Health Sciences, Arba Minch University, Arba Minch, Ethiopia

* habtamuwondmagegn@gmail.com

## Abstract

### Background

Malnutrition among children and adolescents with cancer in low- and middle income countries significantly contributes to several adverse outcomes which have an impact on health-related quality of life and overall survival. This study aimed to assess the magnitude of malnutrition and its associated factors among pediatric cancer patients on chemotherapy.

### Methods

A health institution-based cross-sectional study was conducted among pediatric cancer patients receiving chemotherapy aged from birth to 15 years attending at Black Lion specialized hospital and Saint Paul's millennium medical college from May 1 to July 1, 2024. By using a simple random sampling method, 345 participants were selected for face to face interview and anthropometric assessments. Pre-tested semi-structured questionnaire, chart review and anthropometric measurement were used. Variables with p-value <0.25 in the bi-variable logistic regression analysis were entered and checked for association in a multivariable logistic regression model. The level of statistical significance was declared at the p-value < 0.05.

### Result

In this study, we successfully enrolled 320 pediatric cancer patients, which represents a response rate of 92.8% (320/345). The magnitude of malnutrition, defined

**Data availability statement:** The dataset includes sensitive participant information and cannot be publicly shared due to ethical restrictions approved by the Institutional Review Board (IRB) of Arba Minch University, as well as the terms of informed consent signed by study participants. Public disclosure would compromise participant privacy and confidentiality. However, de-identified data may be made available to qualified researchers upon reasonable request. Such requests will be subject to review and approval by the IRB of Arba Minch University. Data access requests may be directed to the Institutional Review Board (IRB) of the College of Medicine and Health Sciences, Arba Minch University (Email: irb@amu.edu.et).

**Funding:** The author(s) received no specific funding for this work.

**Competing interests:** The authors have declared that no competing interests exist.

**Abbreviations:** BMI, Body Mass Index; ETB, Ethiopian Birr; MUAC, Mid-Upper Arm Circumference; STRONGkids, Screening Tool Risk On Nutritional Status And Growth; TRTs, Treatment-Related Toxicities; TSFT, Triceps Skin-Fold Thickness; WHO, World Health Organization.

by low BMI-for-age or weight-for-height/length z-scores, was 28.4% (91/320) (95% CI: 24.8%−31.3%**).** Additionally, the prevalence of stunting, based on height-for-age, was 30.6% (98/320) (95% CI: 26.5%–34.2%).Children in the age category of 11–15 years (AOR = 2.54, 95% CI; 1.18–5.48), Children's of mothers educational level illiterate, (AOR = 2.20, 95%CI; 1.01–4.75), Children form Households which earn <2000 ETB (AOR = 2.93, 95%CI; 1.14–7.53), Children with a cancer duration of 2–4 years (AOR = 1.34, 95%CI; 1.05–1.71), Children with hematologic malignancy (AOR = 2.18, 95%CI; 1.16–3.81), Children who had co-morbidities (AOR = 1.54, 95% CI; 1.12–2.10) and, Children's who have difficulty of swallowing (AOR = 2.11, 95%CI; 1.22–3.95) were significantly associated factors with being malnourished.

## Conclusion

This study identified malnutrition in 28.4% (91/320) of participants undergoing chemotherapy. Children most at risk were those between 11 and 15 years old, from low-income households, with mothers who had no formal education, Children with a cancer duration of 2–4 years, with hematologic malignancies, with co-morbidities, and with difficulty swallowing.

## Background

Cancer, a complex group of diseases characterized by the uncontrolled growth and spread of abnormal cells, is a significant health concern globally [1].The global burden of cancer underscores its impact on public health, with approximately 10 million new cases and 5 million deaths annually. Pediatric cancer patients face an additional challenge, as malnutrition contributes significantly to morbidity and mortality with a prevalence of malnutrition ranging from 10% to 60% worldwide [2], accounting for about 20% of cancer-related deaths [3]. The impact is particularly pronounced in developing countries, where limited resources and healthcare infrastructure pose additional challenges in the management of pediatric oncology cases [4].

The causes of malnutrition in pediatric cancer patients are complex. The disease itself, in combination with severe cancer treatments such as chemotherapy and radiation, can cause metabolic changes, decreased appetite, and increased dietary requirements [5]. According to emerging research, cancer and its treatments increase metabolism, cytokine release, and disturb normal physiological processes, which heightens the risk of malnutrition [6]. Furthermore, the mental stress associated with cancer diagnosis and treatment can worsen nutritional problems among pediatric patients.

Malnutrition among children and adolescents battling cancer in low- and middle-income countries (LMICs) significantly contributes to several adverse outcomes, including premature discontinuation of therapy, heightened incidence and severity of treatment-related toxicities (TRTs), suboptimal clinical responses, and diminished survival rates [6–9]. These deleterious effects exert a detrimental impact on

health-related quality of life [5] and overall survival [6]. Enhancing nutritional status has been demonstrated to mitigate TRTs such as severe infections and mucositis, while concurrently augmenting the five-year overall survival rate [9]. Therefore, it is imperative to assess the prevalence of malnutrition and elucidate its determinants in pediatric cancer patients to effectively prevent or manage these complications

Many studies employ diverse metrics and assessment approaches, leading to considerable disparity in the reported prevalence of malnutrition among pediatric cancer patients. Despite the recognized significance of nutritional support in enhancing treatment outcomes and mitigating complications, there exists a notable deficiency in empirical research concerning the prevalence and contributing determinants of malnutrition among pediatric cancer patients in the specified study region.

This dearth of comprehensive data hinders the development of tailored interventions and the refinement of treatment protocols [10]. Consequently, the present study was aimed to assess magnitude of malnutrition and its associated factor among pediatrics cancer patient on chemotherapy in the study area.

In Ethiopia there are two public oncology centers. The prevalence of childhood cancer and its associated complications, including malnutrition, underscores the critical need for comprehensive and tailored interventions in pediatric oncology. Factors contributing to malnutrition in this population are diverse, involving both the disease and its treatment. Understanding and addressing these factors are essential to improve the overall well-being and outcomes of pediatric cancer patients.

## Methods and materials

### Study area

The study was conducted in St. Paul's Hospital Millennium Medical college hospital and Tikur Anbesa Specialized Hospital. Tikur Anbessa Specialized Hospital (TASH) is a referral teaching hospital. Established in 1964 offers specialized clinical services and is a significant healthcare institution in Ethiopia with a bed capacity of 800. It is located in Addis Ababa serving a population of over 4 million people. It is one of the few federal public hospitals in Ethiopia where patients can obtain advanced comprehensive treatments for cancer including chemotherapy, radiotherapy, and surgery. Under the haemato-oncology unit, there are over 90 staff members. It, therefore, has a high demand for services. Records indicate that there are approximately 25 new cancer patients every week, with 20 patients admitted into the haemato-oncology ward and about 110 patients attended at the outpatient clinic daily.

St. Paul's hospital is located in Gulele sub city and it is the second largest hospital in Ethiopia. St. Paul's hospital was established by Emperor Haile Selassie in 1969. The medical college was formed in 2007 and the hospital has 350 beds, seeing an annual average of 300,000 patients. It has a catchment population of more than 5 million. The hospital has 1200 clinical and non-clinical staff. The oncology center has three departments which see between18–40 patients per day: medical, hematology and genecology. The hospital has 1200 clinical and non-clinical staff.

Based on the daily and weekly attendance figures, it is estimated that the two centers collectively serve a pool of approximately 1,200−1,500 pediatric oncology patients annually, making them the primary referral centers for pediatric cancer in Ethiopia.

### Study design and period

A health institution based cross-sectional study was conducted from May 1 to July 1, 2024.

### Population

All pediatrics diagnosed with cancer attending care at Tikure Anbesa and St. Paul's hospital were source population and selected pediatrics cancer patients aged 15 years and below attending care in Tikure Anbesa and St. Paul's hospital at the

time of data collection period were study population. However, patients who were seriously ill during the data collection or cognitively impaired were excluded from the study.

## Sample size determination

The sample size was calculated using a single population proportion formula. We used a proportion (p) of 0.449 from a previous study on malnutrition among pediatric cancer patients in Jimma, Ethiopia [11]. To ensure the study's feasibility within the data collection period, a margin of error (d) of 5.5% was used instead of the conventional 5%. The calculation was as follows:

$$n = \frac{\left(Z(1-\frac{\alpha}{2})\right)^2 \times p \times (1-P)}{d^2} = \frac{(1.96^2) \times (0.449 \times 0.551)}{0.055^2} \approx 314.$$

**Where:** n = initial sample size, d = 0.55 (margin of error), p = 0.449 (proportion of malnutrition) Zα/2 = 1.96 (the critical value for 95% confidence level)

After adding a 10% contingency for non-response, the final target sample size was 314 + 31 = **345**

During the data collection period, we successfully enrolled 320 pediatric cancer patients who met the inclusion criteria and provided complete data. This achieved sample size provides a margin of error of ±5.6% for the primary outcome, which was deemed acceptable for this study.

## Sampling techniques and procedure

Tikure Anbesa specialized hospital (TASH) and St Paul's hospital millennium medical college (SPHMMC) was purposely selected for this study because they are the only pediatric oncology centers available in the town. A simple random sampling technique was used to recruit the predetermined sample size (345). Before the selection of study participants, proportion to size allocations of the sample size for the number of pediatric oncology patients in each health institution was done. Finally, sampling frame, the required number (n = 345) of target participant were selected by using simple random sampling method random number generator. (Fig 1)

## Measurements and tools

Weight was measured using a digital scale (SECA), to the nearest 0.1 kg; each participant was asked to remove heavy clothes and the scale was calibrated to zero before and after each measurement.

Height was measured to the nearest 0.1 cm using a stadiometer without shoes, the head of participants at the Frankfurt plane, knees straight and the heels, buttocks, calves and shoulder blades touching the vertical stand of the stadiometer.

BMI was calculated as body weight in kilograms divided by the squared value of body height in meters (kg/m2).

Socio-demographic variables such as maternal education, household income, and birth interval were collected as they are established distal determinants of child nutritional status in the literature.

The Anthro software of the World Health Organization (version 3.0.1; Department of Nutrition, WHO) was used to calculate the nutritional status according to the BMI of up to 5-year-old patients and the WHO Anthro-Plus software (version 1.0.2) for patients aged above 5 years.

For this study, undernutrition (wasting and underweight) was defined using Weight-for-Length/Height and BMI-for-age z-scores, as these are sensitive indicators of acute nutritional status, which is a primary concern in the context of active chemotherapy. Stunting (Height-for-Age z-score < −2 SD) was also assessed as an indicator of chronic malnutrition, and its prevalence is reported separately in the results.

## Operational definitions

**Normal nutrition.** Defined as an age-appropriate z-score between −2 and +1 for Weight-for-Length (0–2 years), Weight-for-Height (2–5 years), or BMI-for-age (5–15 years).

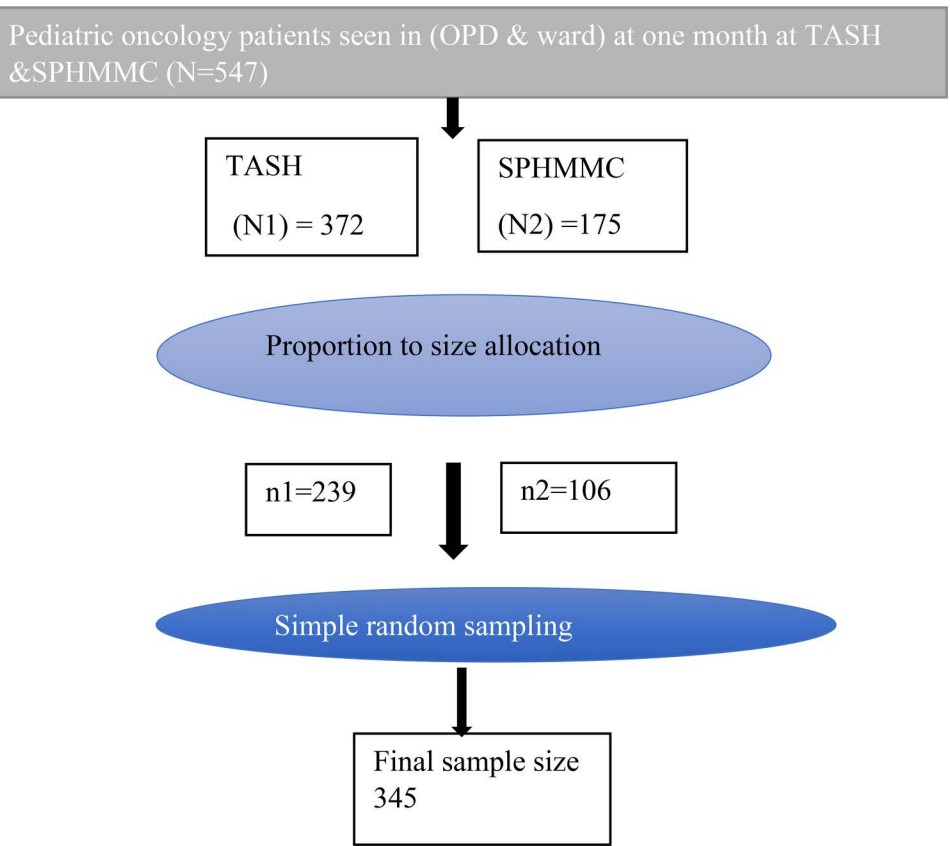

**Fig 1. Schematic representation of the sampling procedure of malnutrition and associated factors among pediatric cancer patients on chemotherapy at TASH & SPHMMMC, Addis Ababa, Ethiopia, 2024.**

**Malnutrition.** Defined as an age-appropriate z-score below −2 for Weight-for-Length (0–2 years), Weight-for-Height (2–5 years), or BMI-for-age (5–15 years).

**Comorbidities** were defined as the presence of any other chronic health condition alongside the cancer diagnosis. These were identified through a review of the patient's medical chart and included conditions such as tuberculosis, HIV, congenital heart disease, and chronic renal disease.

**The duration of cancer** was defined as the time elapsed since the initial diagnosis was confirmed.

**Children.** In this study, the pediatric age group was defined as children aged 15 years and below.

### Data analysis

Data was coded and entered using EpiData v3.1 and analyzed using SPSS v26. Descriptive statistics such as frequency, proportion, mean, and standard deviation were computed. To identify the associated factors, a binary logistic regression model was applied. Both bivariable and multivariable binary logistic regression models were fitted. The model-building process involved entering all independent variables into the bivariable analysis. Variables with a p-value ≤ 0.25 were selected as candidates for the multivariable model. The multivariable model was constructed using a backward stepwise likelihood ratio method to identify factors independently associated with malnutrition. To assess the strength of association, Crude Odds Ratios (COR) and Adjusted Odds Ratios (AOR) with 95% CIs were computed. A significant association was declared at a P-value < 0.05. The goodness-of-fit of the final model was assessed using the Hosmer-Lemeshow test,

and the presence of multicollinearity was evaluated using the Variance Inflation Factor (VIF), with a value of 10 used as a threshold.

## Data quality assurance

To assure the quality of data, properly designed data collection instruments were provided. Training for data collectors and supervisors that include a briefing on the general objective of the study, and discussing the contents of the questionnaire was given. Data collectors and supervisors were trained for 2 days and data collectors were daily supervised by the investigator and supervisor. Pretest was performed on 5% of samples. Findings were used to modify the questionnaire and tool as need. On each day of data collection, data was check for consistency and completeness. The collected data was reviewed and checked for completeness before data entry. The data clearance and variables were coded during data entry.

## Ethical approval and consent to participate

This study was reviewed and approved by the Institutional Review Board (IRB) of St. Paul's Hospital Millennium Medical College (SPHMMC), Addis Ababa, Ethiopia (Reference number: Pm23/1186). Written informed consent was obtained from all parents/guardians of participating children. For children aged 7–15 years, verbal assent was documented in the presence of a witness (a nurse or social worker). The ethics committee waived written consent for children under 7 years, as the study involved no invasive procedures and used anonymized data. The study is conducted according to the principles expressed in the Declaration of Helsinki.

## Result

### Socio-demographic characteristics

During the data collection period, we successfully enrolled 320 pediatric cancer patients, achieving a response rate of 92.8%. The mean ± SD age of the respondents was 6.61 ± 4.1 years and 146(45.6%) were male. The magnitude of malnutrition, defined by low BMI-for-age or weight-for-height/length z-scores, was 28.4% (n = 91/320) (95% CI: 24.8%−31.3%). Additionally, the prevalence of stunting, based on height-for-age, was 30.6% (98/320) (95% CI: 26.5%–34.2%) Mother's educational attainment varied, with 36.6% (117/320) having no formal education, 22.2% having primary education, 22.8% having secondary education, and 17.5% having college or higher education. Average monthly income of families between 2000–4000 ETB were 95(29.6%) and 55% of the participants were rural residents (Table 1).

### Disease and treatment related factors

This study investigated various characteristics of a cancer participant population. Nearly all participants (97.5%) reported no family history of cancer. Comorbidity was present in 52.5% (168/320) of the participants. Solid tumors were the most frequent cancer type (56.6%). The distribution of participants across cancer duration categories (<2 years, 2–4 years, >4 years) and stages (stage 1, 2, 3) was relatively even. Chemotherapy alone was the most common treatment approach (70.6%). The majority of participants reported no issues with chewing or eating (62.8%), and a vast majority (88.5%) had swallowing problems. However, a (22.5%) did experience loss of appetite. (Table 2).

### Magnitude of malnutrition

In the current study, the magnitude of malnutrition, defined by low BMI-for-age or weight-for-height/length z-scores, was 28.4% (91/320) (95% CI: 24.8%−31.3%). Additionally, the prevalence of stunting, based on height-for-age, was 30.6% (98/320) (95% CI: 26.5%–34.2%). Out of the participants who were malnourished 49 (53.8%) were female. The largest percentage of malnourished children (39.6%) falls into the 11–15 age group. (Fig 2)

**Table 1. Socio-demographic characteristics of pediatric cancer patients attending care at SPHMMC and TASH 2024.**

| Variables | Categories | Frequency (n = 320) | Percent (%) |
|---|---|---|---|
| Age of children | <=5 | 144 | 45.0% |
| | 6-10 | 102 | 31.9% |
| | 11-15 | 74 | 23.1% |
| Sex | Female | 174 | 54.4 |
| | Male | 146 | 45.6% |
| Mother's age at pregnancy (years) | 19-25 | 30 | 9.4% |
| | 33-39 | 260 | 81.3% |
| | 42-48 | 30 | 9.4% |
| Mother's Marital status | Married | 215 | 54.4% |
| | Not Married | 105 | 45.6% |
| Mother's educational level | No education | 117 | 36.6% |
| | Primary | 75 | 22.2% |
| | Secondary | 72 | 22.8% |
| | Collage and above | 56 | 17.5% |
| Average monthly income of Family before the pregnancy (ETB) | <2000 | 103 | 36.5% |
| | 2001-4000 | 95 | 29.6% |
| | 4001-6000 | 72 | 22.5% |
| | >6000 | 50 | 15.6% |
| Residence | Rural | 176 | 55.0% |
| | Urban | 144 | 45.0% |
| Birth Order | First | 61 | 19.1% |
| | Second | 101 | 31.6% |
| | Third | 88 | 27.5% |
| | Forth | 42 | 13.1% |
| | Fifth_and_above | 28 | 8.8% |
| Birth interval | <=3 year | 138 | 43.1% |
| | >3year | 182 | 56.9% |
| Number of under 5 children | One | 121 | 37.8% |
| | Two | 71 | 22.2% |
| | Three and above | 128 | 40% |

## Factors associated with malnutrition of pediatric cancer patients on chemotherapy attending at oncology centers of TASH and SPMMC

To identify potential factors associated with malnutrition, a two-step approach was employed. First, bi-variable logistic regression was conducted to screen candidate variables with a p-value less than 0.25. This initial screening helps narrow down the number of variables for further analysis and reduces the risk of overfitting the model.

Following the bi-variable analysis, variables identified as potentially associated with malnutrition (p-value < 0.25) were entered into a multivariable binary logistic regression model. These variables included patient demographics (age, sex, birth order, birth interval), socioeconomic factors (mother's educational level, mother's marital status, residence, average monthly income, number of people in household), cancer characteristics (types of cancer, stage of cancer, number of anticancer drugs), and various treatment side effects experienced in the past month (difficulty of chewing, difficulty of swallowing, diarrhea, loss of appetite, and nausea). This multivariable analysis allows for the exploration of independent associations between each variable and malnutrition while controlling for the influence of other factors in the model.

**Table 2. Disease- and treatment-related factors among pediatrics cancer patients receiving chemotherapy treatment in the cancer center, Ethiopia (n 320).**

| Characteristic | Category | n | (%) |
|---|---|---|---|
| Family history of cancer | No | 312 | 97.5% |
| | Yes | 8 | 2.5% |
| Comorbidity | No | 152 | 47.5% |
| | Yes | 168 | 52.5% |
| Types of cancer | Solid tumor | 181 | 56.6% |
| | Hematologic malignancy | 139 | 43.4% |
| Duration on cancer | < 2 year | 94 | 29.4% |
| | 2-4 year | 133 | 41.6% |
| | >4 year | 93 | 29% |
| Stages of cancer | stage 1 | 94 | 29.4 |
| | stage 2 | 100 | 31.3 |
| | stage 3 | 126 | 39.4 |
| Type of treatment | Chemotherapy_and_radiation | 41 | 12.8% |
| | Chemotherapy and surgery | 53 | 16.6% |
| | chemotherapy alone | 226 | 70.6% |
| Difficulty of chewing | No | 303 | 62.8% |
| | Yes | 17 | 37.2% |
| Difficulty of swallowing | Present | 37 | 11.5.% |
| | Absent | 283 | 88.4% |
| Loss of appetite | No | 248 | 77.5% |
| | Yes | 72 | 22.5% |

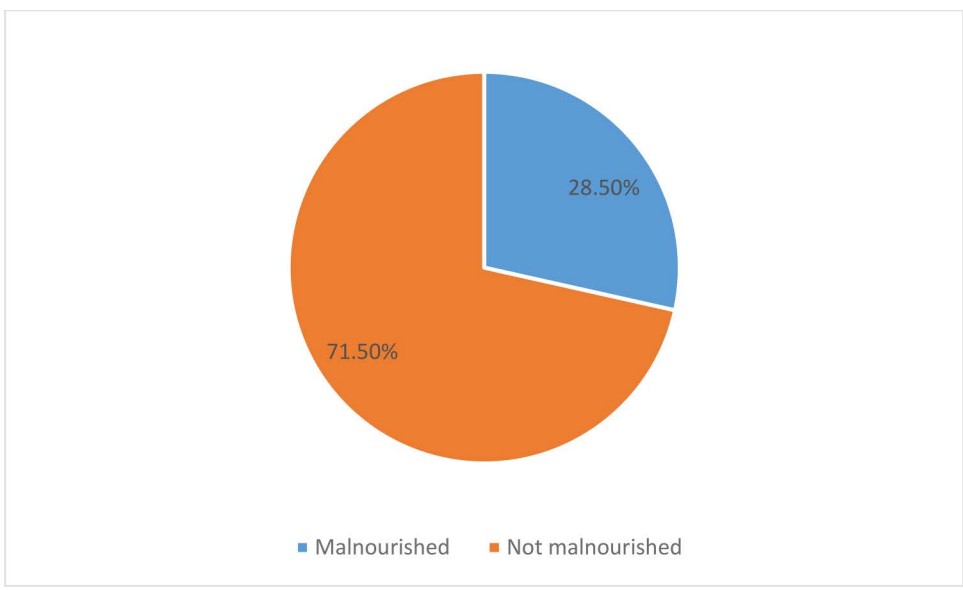

**Fig 2. Magnitude of malnutrition among pediatric cancer on chemotherapy attending at TASH and SPHMC, 2024.**

The results of the multivariable logistic regression will be presented in the following section, highlighting statistically significant factors associated with malnutrition in pediatric cancer patients receiving chemotherapy at TASH and SPMMC oncology centers.

In the multivariable binary logistic regression analysis, several factors were significantly associated with malnutrition: age category (11–15 years), maternal illiteracy, low household income (<2000 ETB), hematologic malignancy, cancer duration (2–4 years), presence of comorbidities, and swallowing difficulty.

Adolescents aged 11–15 years were 2.54 times more likely to be malnourished than children in other age groups (AOR = 2.54, 95% CI: 1.18–5.48). Children of illiterate mothers had 2.20 times higher odds of malnutrition compared to those whose mothers had attained secondary or higher education (AOR = 2.20, 95% CI: 1.01–4.75). Household economic status also played a significant role; children from families earning <2000 ETB per month were 2.93 times more likely to be malnourished than those from households earning >6000 ETB (AOR = 2.93, 95% CI: 1.14–7.53).

Regarding clinical factors, children with hematologic malignancies faced 2.18 times higher odds of malnutrition than those with solid tumors (AOR = 2.18, 95% CI: 1.16–3.81). The presence of comorbidities was also a significant risk factor (AOR = 1.54, 95% CI: 1.12–2.10). Furthermore, children with swallowing difficulties (Present) were 2.11 times more likely to be malnourished than those without such difficulties (Absent) (AOR = 2.11, 95% CI: 1.22–3.95). Finally, a cancer duration of 2–4 years was significantly associated with malnutrition (AOR = 1.34, 95% CI: 1.05–1.71) (Table 3).

## Discussion

This study was conducted to assess the magnitude of malnutrition and associated factors among pediatric oncology patients on chemotherapy in Addis Ababa, Ethiopia. We found a malnutrition prevalence of 28.4% (91/320) based on acute indicators (BMI-for-age or weight-for-height), while the prevalence of stunting was 30.6% (98/320). This high burden of both acute and chronic malnutrition suggests many children in Ethiopia are already nutritionally depleted before they even receive their first dose of medicine. Many arrive at the clinic already physically weakened by long-term nutritional gaps. This baseline vulnerability is then tragically deepened as the cancer drains their remaining energy and the harsh side effects of chemotherapy make it even harder for their young bodies to stay resilient.

The prevalence found in this study (28.4%) is consistent with reports from other low- and middle-income countries (LMICs). A study conducted in Brazil, which assessed the nutritional status of 1,154 children and adolescents diagnosed with malignant neoplasms found 25.8% prevalence of malnutrition [12]. Another study conducted in India found 38% of pediatric cancer patients being malnourished using Body Mass Index (BMI) as the primary measure. However, employing additional anthropometric measurements like triceps skinfold thickness (TSFT) and mid-upper arm circumference (MUAC) yielded higher rates, suggesting the importance of using multiple assessment tools [13]. However, this study found higher prevalence rates than those reported in high-income settings, such as the United States (6%) and Italy (9%) [14,15]. This disparity highlights the "double burden" faced by children in Ethiopia, where the physiological stress of cancer overlaps with factors like poverty, limited access to nutritious food, and inadequate healthcare infrastructure. Developed nations with advanced healthcare systems may offer better nutritional support and proactive interventions, potentially leading to lower malnutrition prevalence.

In this study, children aged 11–15 years had 2.54 times higher odds of malnutrition compared to younger children. This finding is consistent with a study conducted in Korea that found adolescents with cancer had a significantly higher risk of malnutrition compared to younger children [16]. Children are metabolically different than adults and continue to grow and develop during long-term treatment. Adolescents undergo a pubertal growth spurt that significantly increases caloric and protein requirements. When cancer-induced anorexia and chemotherapy-induced nausea are superimposed on these high baseline needs, a nutritional deficit occurs rapidly [17]. In addition, adolescents are more susceptible to depression and body image concerns related to treatment such as alopecia and weight loss, which can lead to "anticipatory nausea" and reduced oral intake. Moreover, older children with cancer may have a longer disease course, increasing their vulnerability to malnutrition.

**Table 3. Factors Associated with malnutrition of pediatric cancer patients on chemotherapy attending at oncology centers of TASH and SPHMMC.**

| Variables | Malnourished (n=91, 28.4%) | Not malnourished(n=229, 71.6%) | COR(95% CI) | AOR(95% CI) | p-value |
|---|---|---|---|---|---|
| **Age (yrs)** | | | | | |
| ≤ 5 | 22(15.3) | 122(84.7) | 1 | 1 | |
| 6–10 | 28(27.5) | 74(72.5) | 1.56(0.92-2.65) | 1.33(0.63-2.82) | .453 |
| 11–15 | 41(55.4) | 33(44.6) | 5.20(2.76- 9.80) | **2.54(1.18-5.48)** | **.017*** |
| **Gender** | | | | | |
| Male | 42(28.8) | 104(71.2) | 1 | **1** | |
| Female | 49(18.2) | 125(71.8) | 1.03(0.21, 0.36) | 1.1(0.91-3.12) | .542 |
| **Birth order** | | | | | |
| < = 3rd | 38 (23.6) | 123(76.4) | 1 | 1 | |
| >3rd | 53(33.3) | 106 (66.7) | 1.40(0.92-2.12) | 0.72(0.35-1.49) | .381 |
| **Birth interval** | | | | | |
| <= 3years | 66(27.7) | 172(72.3) | 2.64(1.53-4.56) | 1.15(0.43-3.07) | .432 |
| >3 years | 25(30.5) | 57(69.5) | 1 | 1 | |
| **Number of under 5 children** | | | | | |
| One | 29(24) | 92(76) | 1 | 1 | .342 |
| Two | 21(29.6) | 50(70.4) | 0.72(0.40-1.30) | 1.1(0.67-3.12) | |
| Three and above | 41(32) | 87(68) | 1.41(0.87-2.28) | 0.23(0.75-3.55) | .261 |
| **Mother's educational level** | | | | | |
| No education | 44(37.6) | 73(62.4) | 4.97(2.20-11.24) | **2.20(1.01-4.75)** | **0.03*** |
| Primary education | 25(33.3) | 50(66.7) | 3.82(1.63-8.99) | 0.87(0.35-2.20) | .771 |
| Secondary | 17(23.6) | 55(76.4) | 2.84(1.16-6.92) | 1.25(0.48-3.24) | .651 |
| College + | 5(8.9) | 51(91.1) | 1 | 1 | |
| **Mother's marital status** | | | | | |
| Married | 54(25.2) | 161(74.8) | 1 | 1 | |
| Not married | 37(35.2) | 68(64.8) | 1.58(0.98-2.54) | 1.31(0.45-8.22) | .226 |
| **Family average monthly income (ETB)** | | | | | |
| ≤ 2000 | 41(39.8) | 62(60.2) | 13.67(4.22-44.23) | **2.93(1.14-7.53)** | **.03*** |
| 2001-4000 | 28(29.5) | 67 (70.5) | 9.33(2.84-30.84) | 0.99(0.44-2.26) | .999 |
| 4001-6000 | 19(26.4) | 53 (73.6) | 6.33(1.83-21.78) | 0.90(0.43-1.90) | .783 |
| >6000 | 3(6) | 47 (94) | 1 | 1 | |
| **Residence** | | | | | |
| Rural | 61(34.7) | 115(65.3) | 2.03(1.28-3.22) | 0.97(0.49-2.05) | .992 |
| Urban | 30(20.8) | 114(79.2) | 1 | 1 | |
| **Duration of cancer** | | | | | |
| <2years | 32(34.1) | 62(65.9) | 1 | 1 | |
| 2-4 years | 42(31.6) | 91(68.4) | 0.92(0.56-1.52) | **1.34(1.05–1.71)** | **.003*** |
| >4 years | 17(18.3) | 76(81.7) | 0.44(0.22-0.87) | 0.54(0.22-1.35) | .212 |
| **Types of cancer** | | | | | |
| Solid tumor | 34(18.9) | 147(81.2) | 1 | 1 | |
| Hematologic malignancy | 57(41) | 82(60) | **2.80**(1.96-4.00) | **2.18(1.16-3.81)** | **.031*** |
| **Co morbidity** | | | | | |
| Yes | 73(43.4) | 95(56.5) | **6.18(3.68-10.42)** | **1.54(1.12–2.10)** | **.018*** |
| No | 18(11.8) | 134(88.2) | 1 | 1 | |

*(Continued)*

**Table 3.** (Continued)

| Variables | Malnourished (n=91, 28.4%) | Not malnour- ished(n=229, 71.6%) | COR(95% CI) | AOR(95% CI) | p-value |
|---|---|---|---|---|---|
| **Difficulty of chewing** | | | | | |
| Yes | 7(41.2) | 10 (58.8) | 0.70(0.44-1.13) | 0.65(0.53-1.78) | .186 |
| No | 84(27.7) | 219(72.3) | 1 | **1** | |
| **Difficulty of swallowing** | | | | | |
| Present | 16 (43.2) | 21(56.8) | **2.04(1.09-3.82)** | **2.11(1.22-3.95)** | **.043*** |
| Absent | 75(26.5) | 208(73.5) | 1 | **1** | |
| **Loss of appetite** | | | | | |
| Yes | 25(34.7) | 47(65.3) | **2.13**(1.30-3.49) | 1.56(0.82-5.22) | .221 |
| No | 66(26.6) | 182 (73.4) | 1 | 1 | |
| **Nausea in the past month** | | | | | |
| Yes | 52 (43.7) | 67 (56.3) | **2.83**(1.85-4.33) | 1.98(0.99-4.33) | .342 |
| No | 39 (19.4) | 162(80.6) | 1 | 1 | |

P-value with "*" shows statistically significant variables.

Our result showed a profound impact of "poverty-malnutrition-cancer" triad. Children of illiterate mothers and those from households earning <2000 ETB monthly were significantly more at risk. This finding is supported by a study conducted in in Kenya where they found that children with mothers who had no formal education were more likely to be malnourished [18]. This association was attributed to limited knowledge about nutrition and challenges accessing healthy food. A review done in Italy highlighted the global link between lower maternal education and child malnutrition in chronic illnesses, including cancer [19]. Moreover, socioeconomic constraints related to education potentially impacting food security and access to nutritious options.

Children from households with a monthly income below 2000 ETB were significantly more likely to be malnourished compared to those from wealthier families. This finding is supported in a study in Bangladesh found a significant association between lower socioeconomic status and malnutrition in children with cancer [20]. A review conducted in India emphasized the well-established link between poverty and malnutrition globally [21]. Lower income families often struggle to afford nutritious food or fresh produce. Limited financial resources force a "trade-off" where families may prioritize the cost of transportation to the hospital over the purchase of high-protein, calorie-dense foods (like meat, eggs, or dairy) necessary for a child on chemotherapy. This creates a cycle where poverty leads to malnutrition, which in turn increases treatment toxicity and hospital stay costs.

Children with a cancer duration of 2–4 years had 1.34 times the odds of being malnourished. A study in India observed increased malnutrition risk in children with cancer who had been in treatment for longer durations [6]. This could be due to the cumulative effects of treatment side effects, where chronic low-grade nausea and altered taste (dysgeusia) lead to a slow but steady decline in nutritional reserves. In contrast, some studies, like one by in Korea, suggest a higher risk for malnutrition during the initial diagnosis and treatment phases [22]. More research is needed to understand the specific mechanisms behind the association between cancer duration and malnutrition, considering factors like type of cancer and specific treatment regimens.

This study found that children with hematologic malignances were more likely to be malnourished than those with solid tumors. This finding contrasts with a study from Malaysia, which reported a higher prevalence of malnutrition in children with solid tumors, possibly due to associated symptoms like bowel obstructions, nausea, and vomiting, or differences in treatment regimens [22]. Finally, children with swallowing difficulties were significantly more likely to be malnourished. This is a direct consequence of the physical challenge of ingesting food and can lead to inadequate calorie and nutrient intake.

In pediatric oncology, this is often a clinical manifestation of chemotherapy-induced mucositis or secondary fungal infections like oral candidiasis due to immunosuppression.

The choice of anthropometric tools influences the reported prevalence of malnutrition. Our study primarily used BMI-for-age and weight-for-height to define malnutrition, focusing on acute weight loss and wasting, which are critical in acutely ill patients. This may underrepresent children with sarcopenia or fluid retention. The high prevalence of stunting we observed suggests a significant burden of chronic malnutrition, which may predate the cancer diagnosis. Future studies in this population would benefit from incorporating composite tools like MUAC and TSFT, or functional assessments, to provide a more comprehensive nutritional picture.

### Limitation of the study

One notable limitation is the cross-sectional design, which captures a snapshot of malnutrition prevalence at a single point in time but does not account for changes over the course of treatment. This design limits the ability to observe the progression of nutritional status and its impact on treatment outcomes, longitudinally. Additionally, the study relies on anthropometric and not with biochemical measurements, which, while comprehensive, may not fully capture the complex and multifactorial nature of malnutrition, especially in the context of varying cancer types and treatment regimens. Finally, the study's findings may not be entirely generalizable to all pediatric oncology populations, particularly those in regions not represented in the sample.

Furthermore, the cross-sectional nature of this study limits the ability to establish causality and to distinguish whether the observed malnutrition was pre-existing, a consequence of the cancer itself, or a result of chemotherapy treatment.

### Conclusions

In conclusion, this study demonstrates a high prevalence of malnutrition (28.4%) among pediatric cancer patients undergoing chemotherapy in Addis Ababa. Adolescents (11–15 years), children from households facing extreme poverty, those with hematologic malignancies, and those with swallowing difficulties were identified as being at the highest risk. These findings highlight the urgent necessity for routine nutritional screening and integrated support interventions in pediatric oncology centers to improve treatment efficacy and survival.

### Acknowledgments

The authors extend their gratitude to the healthcare professionals and data collectors for their invaluable assistance in collecting the data.

### Author contributions

**Conceptualization:** Habtamu Wondmagegn Atlaw.

**Data curation:** Tsegazeab Ayele Meshesha, Maycas Gembe, Habtamu Esubalew Bezie.

**Formal analysis:** Sayih Mehari Degualem.

**Methodology:** Habtamu Wondmagegn Atlaw, Biniyam Demisse Andarge, Habtamu Esubalew Bezie.

**Project administration:** Edlework Wondmagegn Atlaw.

**Supervision:** Habtamu Wondmagegn Atlaw.

**Writing – original draft:** Edlework Wondmagegn Atlaw, Maycas Gembe.

**Writing – review & editing:** Biniyam Demisse Andarge, Sayih Mehari Degualem, Tsegazeab Ayele Meshesha.

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
