## [Decision Letter · Decision Letter 0]

27 Oct 2025

Dear Dr. Atlaw,

Thank you for submitting your manuscript to PLOS ONE. After careful consideration, we feel that it has merit but does not fully meet PLOS ONE’s publication criteria as it currently stands. Therefore, we invite you to submit a revised version of the manuscript that addresses the points raised during the review process.

We look forward to receiving your revised manuscript.

Kind regards,

Syeda Humaida Hasan, Diploma in Child Health, FCPS (Pediatrics)

Academic Editor

PLOS ONE

Journal Requirements:

2. We note that your Data Availability Statement is currently as follows: All relevant data are within the manuscript and in Supporting Information files.

3. Please include a caption for figure 2.

Additional Editor Comments:

In addition to the reviewers’ comments, please address the following points to improve clarity and methodological transparency:

Sample size calculation: Clarify why the “proportion of pregnancy within 12 months” was used in the sample size estimation. Explain its relevance to the study outcome and provide a supporting reference.

Duration of cancer: The inclusion criterion of “duration of cancer less than 2 years” appears to include newly diagnosed cases (<1 month). In such cases, malnutrition may result from the malignancy itself rather than its treatment or chronic course. Please revise or justify this criterion in line with your study objectives.

Tables and demographic variables: Tables should be formatted more clearly. For example, “marital status” should be specified as “marital status of the mother.” whereas the inclusion of “birth interval” is not clearly useful. If you choose to retain it, please clarify its relevance to the study.

Please address these points along with the reviewers’ comments in your revised manuscript and response letter.

Best regards,

Dr. Syeda Humaida Hasan

Academic Editor

Reviewers' comments:

Reviewer's Responses to Questions

**Comments to the Author**

1. Is the manuscript technically sound, and do the data support the conclusions?

Reviewer #1: No

Reviewer #2: No

2. Has the statistical analysis been performed appropriately and rigorously?

Reviewer #1: No

Reviewer #2: Yes

3. Have the authors made all data underlying the findings in their manuscript fully available?

Reviewer #1: Yes

Reviewer #2: Yes

4. Is the manuscript presented in an intelligible fashion and written in standard English?

Reviewer #1: Yes

Reviewer #2: No

Reviewer #1: In this paper, the Authors analyze and report key contributors to pediatric patients' malnutrition in low-and mid- income countries. They identify factors associated with malnutrition, including age of the children, education level of the mothers, family income, duration of cancer, cancer types, and so on. The Authors also quantify the association of these factors with magnitude of malnutrition. This study provide valuable information.

However there are several major issues the Authors need to address.

Line 123: the equation is difficult to understand. Please explain the variables with more details. Apparently the P and p represent the same variable, and the meaning of it on line 125 does not make sense to me. Furthermore, the result of the equation does not equal to the result of the Authors.

Line 364: many of these Abrevations are not referenced in the main text. In addition some concepts need to be explained, e.g., AOR and COR.

In general, the Authors should provide more technical details on how the models were fitted.

Reviewer #2: The authors present an assessment of malnutrition in pediatric oncology patient in Ethiopia, and associations with several factors.

My main comments are :

1-As of today, there is still no consensus on the best definition of pediatric malnutrition. WHO defines malnutrition as including undernutrition (stunting and wasting), overweight, and micronutrient deficiencies. The authors use Weight-for-Length / weight-for-height z-score = wasting and BMI for age z-score . This is done without justification. Why do they use only these parameters? Based on what reference/guidelines? It should be argued. Why don't they assess stunting (height-for-age) as well?

This should be made clear in the methods sections and choices should be clearly explained in the discussion, which is not. Also, it would be usefull to present separately the rate of wasting vs. stunting vs. low BMI in the cohort. Parameters should be identified in the abstract, instead of simply refering as 'malnutrition'.

In the discussion, they should argue their choice of tools to define malnutrition and their limits.

2- Authors present that presence of comorbidities was associated with malnutrition. These comorbidities are not defined in the methods sections. What are those comorbidities? How were they identified/defined?

3- Writing and English must be revised. They are too many sentences to state here. For example: ‘’Children with 2-4 years duration of cancer were significantly associated with being malnourished. ‘’ Syntax must be revised: The children were not associated with malnutrition!

Minor comments:

4- ''Children form Households which earn <2000 a month'' : precise the currency

5- Methods: the yearly pool of pediatric cancer patients should be given for each center. In my understanding, authors only indicated the number of cancer cases (in total).

**Do you want your identity to be public for this peer review?** For information about this choice, including consent withdrawal, please see our Privacy Policy

Reviewer #1: No

Reviewer #2: **Yes:** Valerie Marcil

---

## [Author Response · Author response to Decision Letter 1]

20 Nov 2025

Response to Reviewers

To: Syeda Humaida Hasan, Diploma in Child Health, FCPS (Pediatrics), Academic Editor, PLOS ONE, and the Esteemed Reviewers

From: Habtamu Wondmagegn (Corresponding Author)

Re: Revision of Manuscript PONE-D-25-44535 – "Magnitude of Malnutrition and Its Associated Factors among Pediatric Cancer Patients on Chemotherapy at Oncology Centers in Addis Ababa, Ethiopia, 2024."

Dear Dr. Hasan and Reviewers,

We would like sincerely thank you for the constructive feedback and the opportunity to revise our manuscript. The comments we have received have been helpful to improve the quality, clarity, and scientific rigor of our work. We have carefully addressed each point raised, and the changes are detailed in a point-by-point manner below.

All revisions in the manuscript are highlighted in the 'Revised Manuscript with Track Changes' file. We believe the manuscript has been greatly improved and now fully meets the publication standards of PLOS ONE.

Response to Journal Requirements

1. Style Requirements: We have reformatted the entire manuscript to comply with the PLOS ONE style templates.

2. Data Availability: The dataset includes sensitive participant information and cannot be publicly shared due to ethical restrictions approved by the Institutional Review Board (IRB) of Arba Minch University, as well as the terms of informed consent signed by study participants. Public disclosure would compromise participant privacy and confidentiality. However, de-identified data may be made available to qualified researchers upon reasonable request. Such requests will be subject to review and approval by the IRB of Arba Minch University. Data access requests may be directed to the Institutional Review Board (IRB) of the College of Medicine and Health Sciences, Arba Minch University (Email: irb@amu.edu.et).

3. Figure Caption: A descriptive caption has been added for Figure 2.

4. Citation of Recommended Works: We have reviewed the literature suggested by the reviewers and have incorporated relevant citations to improve the context and depth of our discussion.

Response to Additional Editor Comments

Comment 1: Sample size calculation: Clarify why the “proportion of pregnancy within 12 months” was used in the sample size estimation. Explain its relevance to the study outcome and provide a supporting reference.

Response: We apologize for this confusing error. The mention of "proportion of pregnancy" was a mistake during the write-up and is entirely irrelevant to our study. We have removed it in the revised version. The sample size was correctly calculated based on the proportion of malnutrition (p=0.449) from the referenced Jimma study (Ref 11). We have made revisions the 'Sample size determination' section for accuracy and clarity, as detailed in our response to Reviewer #1.

Comment 2: Duration of cancer: The inclusion criterion of “duration of cancer less than 2 years” appears to include newly diagnosed cases (<1 month). In such cases, malnutrition may result from the malignancy itself rather than its treatment or chronic course. Please revise or justify this criterion in line with your study objectives.

Response: This is a very insightful point. We have clarified in the 'Methods' section (lines 178-179) that "duration of cancer" refers to the "time since diagnosis." We agree that in cross-sectional studies like ours, it is challenging to determine whether malnutrition is a consequence of the pre-disease status, the cancer itself, or the treatment. To specifically address this, we added a sentence to the 'Limitations' section (lines 383-385): "Furthermore, the cross-sectional nature of this study limits the ability to establish causality and to distinguish whether the observed malnutrition was pre-existing, a consequence of the cancer itself, or a result of chemotherapy treatment."

Comment 3: Tables and demographic variables: Tables should be formatted more clearly. For example, “marital status” should be specified as “marital status of the mother.” whereas the inclusion of “birth interval” is not clearly useful. If you choose to retain it, please clarify its relevance to the study.

Response: We appreciate the editor's recommendation.

• We have revised Table 1 for clarity. "Marital status" is now specified as "Mother's marital status."

• Since "birth interval" is a known demographic feature linked to child nutrition in our scenario, we have kept it. We have added a brief justification for its inclusion in the 'Methods' section under 'Data Collection' (lines 155-156): "Socio-demographic variables such as maternal education, household income, and birth interval were collected as they are established distal determinants of child nutritional status in the literature."

Response to Reviewers' Comments

Reviewer 1

We sincerely thank Reviewer #1 for their valuable and technical feedback, which has greatly improved the methodological transparency of our manuscript.

Comment 1 (Line 123): The equation is difficult to understand. Please explain the variables with more details. Apparently the P and p represent the same variable, and the meaning of it on line 125 does not make sense to me. Furthermore, the result of the equation does not equal to the result of the Authors.

Response: We sincerely apologize for these errors and the lack of clarity. We have completely rewritten this section.

• We have standardized the variable for proportion to p and clearly defined all variables (n, Zα/2, p, d) below the formula.

• We have corrected the calculation. The initial calculation of 301 was incorrect. The revised calculation is (1.96) ² * (0.449 * 0.551) / (0.055) ² ≈ 314. With a 10% non-response rate, the target was 345. We enrolled 320 participants, which provides a margin of error of ±5.6%, which is still precise and acceptable. Please see the revised 'Sample size determination' section (lines 120-133).

Comment 2 (Line 364): Many of these Abbreviations are not referenced in the main text. In addition, some concepts need to be explained, e.g., AOR and COR.

Response: Thank you for pointing this out.

• After going over the list of abbreviations, we eliminated those that weren't used in the paper (such as AML, BL, HL, HFA, and WFH). Please see this at (line 396 – 403).

• We have defined AOR (Adjusted Odds Ratio) and COR (Crude Odds Ratio) upon their first use in the 'Data analysis' section (lines 190-191): "To assess the strength of association, Crude Odds Ratios (COR) and Adjusted Odds Ratios (AOR) with 95% CIs were computed."

Comment 3: In general, the Authors should provide more technical details on how the models were fitted.

Response: We have expanded the 'Data analysis' section (lines 182-194) to provide more detail:

"Both bivariable and multivariable binary logistic regression models were fitted. The model-building process involved entering all independent variables into the bivariable analysis. Variables with a p-value < 0.25 in the bivariable analysis were selected as candidates for the multivariable model. The multivariable model was then constructed using a backward stepwise likelihood ratio method to identify factors independently associated with malnutrition. The goodness-of-fit of the final model was assessed using the Hosmer-Lemeshow test, and the presence of multicollinearity among the independent variables was evaluated using the Variance Inflation Factor (VIF), with a value of 10 used as a threshold for serious multicollinearity."

Reviewer 2

We appreciate Dr Valérie Marcil's insightful and helpful feedback, which greatly improved our manuscript's conceptual framework and clarity.

Comment 1: As of today, there is still no consensus on the best definition of pediatric malnutrition... Why do they use only these parameters?... Why don't they assess stunting (height-for-age) as well?... Parameters should be identified in the abstract... argue their choice of tools...

Response: This is a critical point, and we have substantially revised the manuscript to address it. We now revised and added a clarification in the methods, result, abstract and result section on the revised manuscript.

• Methods: We have added a justification in the 'Measurements' section (lines 158-165): "Nutritional status was assessed using WHO Anthro software. For this study, undernutrition (wasting and underweight) was defined using Weight-for-Length/Height and BMI-for-age z-scores, as these are sensitive indicators of acute nutritional status, which is a primary concern in the context of active chemotherapy. Stunting (Height-for-Age z-score < -2 SD) was also assessed as an indicator of chronic malnutrition, and its prevalence is reported separately in the results."

• Results: We have added the prevalence of stunting to the 'Magnitude of malnutrition' section (line 219-221): "The prevalence of wasting/low BMI-for-age was 28.5%. Additionally, the prevalence of stunting, based on height-for-age, was 30.5%."

• Abstract: We have specified the parameters used (line 34-36): "...the magnitude of malnutrition, defined by low BMI-for-age or weight-for-height/length z-scores..."

• Discussion: We have added a new paragraph (lines 365-371) discussing our choice of tools and their limitations: "The choice of anthropometric tools influences the reported prevalence of malnutrition. Our study primarily used BMI-for-age and weight-for-height to define malnutrition, focusing on acute weight loss and wasting, which are critical in acutely ill patients. This may underrepresent children with sarcopenia or fluid retention. The high prevalence of stunting we observed suggests a significant burden of chronic malnutrition, which may predate the cancer diagnosis. Future studies in this population would benefit from incorporating composite tools like MUAC and TSFT, or functional assessments, to provide a more comprehensive nutritional picture."

Comment 2: Authors present that presence of comorbidities was associated with malnutrition. These comorbidities are not defined in the methods sections. What are those comorbidities? How were they identified/defined?

Response: Thank you. In the revised manuscript we defined "comorbidities" in the 'Methods' section under 'Data Collection' (lines 174–177): "Comorbidities were defined as any other chronic health condition that coexisted with the cancer diagnosis. A study of the patient's medical chart revealed illnesses such as tuberculosis, HIV, congenital heart disease, and chronic kidney failure”.

Comment 3: Writing and English must be revised. They are too many sentences to state here. For example: ‘’Children with 2-4 years’ duration of cancer were significantly associated with being malnourished.’’

Response: We thoroughly review the entire article to remove grammatical faults and increase clarity. The sentence highlighted has been corrected to: "A cancer duration of 2-4 years was significantly associated with malnutrition." (Line 261, 269,388.,). We have also engaged a colleague proficient in academic English to conduct a final evaluation.

Comment 4: ''Children form Households which earn <2000 a month'': precise the currency

Response: We have added "ETB" (Ethiopian Birr) throughout the manuscript where income is mentioned (e.g., Abstract, Results, Tables 1 & 3).

Comment 5: Methods: the yearly pool of pediatric cancer patients should be given for each center.

Response: We have added an estimate to the 'Study Area' section (lines 108-110): "Based on the daily and weekly attendance figures, it is estimated that the two centers collectively serve a pool of approximately 1,200-1,500 pediatric oncology patients annually, making them the primary referral centers for pediatric cancer in Ethiopia."

Conclusion

We really appreciate the editor and reviewers' effort and expertise. We feel that by addressing all of the issues mentioned, we have considerably improved the quality and impact of our paper. We hope that the amended version is now considered appropriate for publishing in PLOS ONE.

Sincerely,

Habtamu Wondmagegn, MSc

On behalf of all co-authors.

---

## [Decision Letter · Decision Letter 1]

1 Jan 2026

Dear Dr. Atlaw,

Thank you for submitting your manuscript to PLOS ONE. After careful consideration, we feel that it has merit but does not fully meet PLOS ONE’s publication criteria as it currently stands. Therefore, we invite you to submit a revised version of the manuscript that addresses the points raised during the review process.

Although both reviewers have indicated that their comments were addressed, further editorial assessment has identified several issues that still require correction. Therefore, the manuscript will require additional revision before it can be considered for further processing. The authors are requested to address the following points carefully:

Please re-write the operational definition of normal nutrition and malnutrition with clarity.

If a 95% confidence interval is used for weight-for-height, the same should also be reported for height-for-age. Please ensure consistency by correcting this in the Results section, Abstract, and Discussion.

It is preferable to report absolute numbers alongside percentages throughout the manuscript. Please revise accordingly.

Please rewrite lines 239–240 and 267–279. The language is repetitive and monotonous, and there is scope for improvement in writing style, variation in sentence structure, and clarity. Additionally, please review punctuation and capitalization carefully, particularly around line 339.

The Conclusion section should be rewritten in clear, standard English to better reflect the study findings.

In the tables:

Please change “age of a children” to “age of children.”

Correct the use of the apostrophe in “mothers’ marital status.”

In the logistic regression analysis, please explain why “no swallowing difficulty” was not assessed against “swallowing difficulty” as the reference category. This requires clarification in the Methods or Results section.

In the Discussion section, please ensure that all key findings are adequately explained and compared with results from similar studies. This section still requires further improvement in depth and interpretation.

Please note that data availability is a prerequisite for PLOS publications. As previously requested, the authors must share an anonymized dataset after obtaining permission from the appropriate authority and include a proper Data Availability statement.

Overall, the manuscript still requires improvement in scientific writing, clarity, and compliance with journal requirements. We encourage the authors to revise the manuscript thoroughly before resubmission.

Thank you.

We look forward to receiving your revised manuscript.

Kind regards,

Syeda Humaida Hasan, Diploma in Child Health, FCPS (Pediatrics)

Academic Editor

PLOS One

Journal Requirements:

Reviewers' comments:

Reviewer's Responses to Questions

**Comments to the Author**

Reviewer #1: All comments have been addressed

Reviewer #2: All comments have been addressed

2. Is the manuscript technically sound, and do the data support the conclusions?

Reviewer #1: Yes

Reviewer #2: Yes

3. Has the statistical analysis been performed appropriately and rigorously?

Reviewer #1: Yes

Reviewer #2: Yes

4. Have the authors made all data underlying the findings in their manuscript fully available?

Reviewer #1: Yes

Reviewer #2: Yes

5. Is the manuscript presented in an intelligible fashion and written in standard English?

Reviewer #1: Yes

Reviewer #2: Yes

Reviewer #1: (No Response)

Reviewer #2: The authors have adressed my comments in a satisfactory manner. I believe the manuscript is now suitable for publication.

**Do you want your identity to be public for this peer review?** For information about this choice, including consent withdrawal, please see our Privacy Policy

Reviewer #1: No

Reviewer #2: **Yes:** Valerie Marcil

---

## [Author Response · Author response to Decision Letter 2]

14 Jan 2026

Response to Reviewers

To: Syeda Humaida Hasan, Diploma in Child Health, FCPS (Pediatrics), Academic Editor, PLOS ONE

From: Habtamu Wondmagegn (Corresponding Author)

Re: Revision of Manuscript PONE-D-25-44535 – "Magnitude of Malnutrition and Its Associated Factors among Pediatric Cancer Patients on Chemotherapy at Oncology Centers in Addis Ababa, Ethiopia, 2024."

Dear Dr. Hasan,

We would like sincerely thank you for the constructive feedback and the opportunity to revise our manuscript. The comments we have received have been helpful to improve the quality, clarity, and scientific rigor of our work. We have carefully addressed each point raised, and the changes are detailed in a point-by-point manner below.

All revisions in the manuscript are highlighted in the 'Revised Manuscript with Track Changes' file. We believe the manuscript has been greatly improved and now fully meets the publication standards of PLOS ONE.

Editor’s Point 1. Please re-write the operational definition of normal nutrition and malnutrition with clarity.

Response: Thank you, the definitions have been clarified in the Operational Definitions section to specify that status is determined based on age-appropriate indicators. Please find this on line number 164-168.

Editor’s Point 2. If a 95% confidence interval is used for weight-for-height, the same should also be reported for height-for-age. Please ensure consistency by correcting this in the Results section, Abstract, and Discussion.

Response: Thank you, the 95% confidence interval for stunting (height-for-age) has been calculated and added to the Abstract, Results, and Discussion sections. Please find this on line number 35, 217,236, and 283

Editor’s Point 3. It is preferable to report absolute numbers alongside percentages throughout the manuscript. Please revise accordingly.

Response: Thank you, absolute numbers (n/N) have been added alongside percentages in the text. For example: The magnitude of malnutrition was 28.4% (n=91/320). The prevalence of stunting was 30.6% (n=98/320). Please find this on line number 235, 236, 282, and 283.

Editor’s Point 4. Please rewrite lines 239–240 and 267–279. The language is repetitive and monotonous, and there is scope for improvement in writing style, variation in sentence structure, and clarity. Additionally, please review punctuation and capitalization carefully, particularly around line 339.

Response: Thank you, we have thoroughly revised the indicated sections to improve the writing style, vary sentence structure, and enhance clarity. We have eliminated monotonous phrasing and ensured a more professional, academic tone. Additionally, we have carefully reviewed the entire manuscript for punctuation and capitalization errors. Please find this on line number 235- 238, 260 – 277.

Editor’s Point 5. The Conclusion section should be rewritten in clear, standard English to better reflect the study findings.

Response: Thank you for your comment. The conclusion has been refined for clarity and impact: as follows;” In conclusion, this study demonstrates a high prevalence of malnutrition (28.4%) among pediatric cancer patients undergoing chemotherapy in Addis Ababa. Adolescents (11–15 years), children from households facing extreme poverty, those with hematologic malignancies, and those with swallowing difficulties were identified as being at the highest risk. These findings highlight the urgent necessity for routine nutritional screening and integrated support interventions in pediatric oncology centers to improve treatment efficacy and survival.”

In the tables:

Editor’s Point 6. Please change “age of a children” to “age of children.”

Response: Thank you, we have updated from "Age of a children" to "Age of children".

Editor’s Point 7. Correct the use of the apostrophe in “mothers’ marital status.”

Response: Thank you, we have updated from“mothers’ marital status’’ to “mother’s marital status.”

Editor’s Point 8. In the logistic regression analysis, please explain why “no swallowing difficulty” was not assessed against “swallowing difficulty” as the reference category. This requires clarification in the Methods or Results section.

Response: We are very grateful to the Editor for identifying this discrepancy. Upon thorough review, we found that the rows for this variable in Table 3 were completely mislabeled in the previous version. The labels "Yes" and "No" (now updated to "Present" and "Absent") were inadvertently swapped, and the reference category was incorrectly assigned in the table layout.

We have corrected Table 3 in the revised manuscript. We now use "Absent" (no swallowing difficulty) as the reference category (AOR = 1.0). Consequently, the data now correctly shows that children with the "Present" status for swallowing difficulty have a significantly higher risk of malnutrition (AOR = 2.11, 95\% CI: 1.22–3.95), which is consistent with our results text, discussion, and clinical logic.

Editor’s Point 9. In the Discussion section, please ensure that all key findings are adequately explained and compared with results from similar studies. This section still requires further improvement in depth and interpretation.

Response: Thank you, we have extensively revised the Discussion section to provide greater scientific depth and clinical interpretation. Rather than simply listing comparative prevalence rates, we have now contextualized our findings within the unique socioeconomic and clinical landscape of Ethiopia. Please find this in line number 280-348

Editor’s Point 10. Please note that data availability is a prerequisite for PLOS publications. As previously requested, the authors must share an anonymized dataset after obtaining permission from the appropriate authority and include a proper Data Availability statement.

Response: We have updated our Data Availability statement to comply with PLOS ONE requirements. Ethical clearance for this study was obtained from the Institutional Review Board (IRB) of Saint Paul’s Millennium Medical College (SPHMMC). Because the dataset contains sensitive clinical information regarding pediatric oncology patients, the data are available upon reasonable request from the SPHMMC IRB for researchers who meet the criteria for access to confidential data. Please find this in line number 386- 390.

Interested researchers may contact the IRB via email at: irb@sphmmc.edu.et.

---

## [Decision Letter · Decision Letter 2]

27 Jan 2026

Magnitude of Malnutrition and Its Associated Factors among Pediatric Cancer Patients on Chemotherapy at Oncology Centers in Addis Ababa, Ethiopia, 2024.

PONE-D-25-44535R2

Dear Dr. Habtamu,

We’re pleased to inform you that your manuscript has been judged scientifically suitable for publication and will be formally accepted for publication once it meets all outstanding technical requirements.

Kind regards,

Kahsu Gebrekidan, Ph.D.

Academic Editor

PLOS One

Additional Editor Comments (optional):

Reviewers' comments:

Reviewer's Responses to Questions

**Comments to the Author**

Reviewer #1: All comments have been addressed

Reviewer #2: All comments have been addressed

2. Is the manuscript technically sound, and do the data support the conclusions?

Reviewer #1: Yes

Reviewer #2: Yes

3. Has the statistical analysis been performed appropriately and rigorously?

Reviewer #1: Yes

Reviewer #2: Yes

4. Have the authors made all data underlying the findings in their manuscript fully available?

Reviewer #1: Yes

Reviewer #2: Yes

5. Is the manuscript presented in an intelligible fashion and written in standard English?

Reviewer #1: Yes

Reviewer #2: Yes

Reviewer #1: (No Response)

Reviewer #2: Following this second revision, all comments were addressed. I believe that the manuscript is suitable for publication.

**Do you want your identity to be public for this peer review?** For information about this choice, including consent withdrawal, please see our Privacy Policy

Reviewer #1: No

Reviewer #2: **Yes:** Valerie Marcil

---

## [Editor Report · Acceptance letter]

PONE-D-25-44535R2

PLOS One

Dear Dr. Atlaw,

I'm pleased to inform you that your manuscript has been deemed suitable for publication in PLOS One. Congratulations! Your manuscript is now being handed over to our production team.

Kind regards,

on behalf of

Dr. Kahsu Gebrekidan

Academic Editor

PLOS One